# Indoxyl Sulfate Alters the Humoral Response of the ChAdOx1 COVID-19 Vaccine in Hemodialysis Patients

**DOI:** 10.3390/vaccines10091378

**Published:** 2022-08-24

**Authors:** Yi-Chou Hou, Chia-Lin Wu, Kuo-Cheng Lu, Ko-Lin Kuo

**Affiliations:** 1Division of Nephrology, Department of Medicine, Cardinal-Tien Hospital, New Taipei City 231, Taiwan; 2School of Medicine, Fu Jen Catholic University, New Taipei City 242, Taiwan; 3Division of Nephrology, Department of Internal Medicine, Changhua Christian Hospital, Changhua 500, Taiwan; 4School of Medicine, Chung-Shan Medical University, Taichung 402, Taiwan; 5Division of Nephrology, Department of Medicine, Taipei Tzu Chi Hospital, Buddhist Tzu Chi Medical Foundation, New Taipei City 231, Taiwan; 6School of Medicine, Buddhist Tzu Chi University, Hualien 970, Taiwan; 7Division of Nephrology, Department of Medicine, Fu Jen Catholic University Hospital, New Taipei City 242, Taiwan; 8School of Post-Baccalaureate Chinese Medicine, Tzu Chi University, Hualien 970, Taiwan

**Keywords:** ChAdOx1, COVID-19 vaccine, end-stage renal disease, hemodialysis, indoxyl sulfate

## Abstract

Background and aims: Vaccination for severe acute respiratory syndrome coronavirus 2(SARS-CoV-2) is strongly recommended. The efficacy of SARS-CoV-2 vaccine for patients with end-stage renal disease is low. Indoxyl sulfate (IS) is a representative protein bound uremic toxin arousing immune dysfunction in CKD patients. It is unknown whether IS impairs the efficacy of vaccines for SARS-CoV-2. Materials and Methods: From 1 June 2021, to 31 December 2021, hemodialysis patients (*n* = 358) and a control group (*n* = 59) were eligible to receive the first dose of the ChAdOx1 COVID-19 vaccine. Titer measurements indicative of the humoral response (anti-S1 IgG and surrogate virus neutralization test (sVNT) results) and indoxyl sulfate concentration measurement were performed 4 weeks after ChAdOx1 vaccine injection. Results: The serum concentrations of anti-S1 IgG were 272 ± 1726 AU/mL and 2111 ± 4424 AU/mL in hemodialysis patients and control group (*p* < 0.05), respectively. The sVNT values were 26.8 ± 21.1% and 54.0 ± 20.2% in the hemodialysis and control groups (*p* < 0.05), respectively. There was a decreasing trend for the anti-S1 IgG titer from the lowest to highest quartile of IS (*p* < 0.001). The patients with higher concentrations of IS had lower sVNT (*p* for trend < 0.001). Conclusion: Hemodialysis patients had weaker humoral immunity after the first dose of the ChAdOx1 vaccine. Higher concentration of IS altered the development of anti-S1 antibodies and sVNT-measured neutralization.

## 1. Introduction

The coronavirus 2019 disease (COVID-19) pandemic was initiated by severe acute respiratory syndrome coronavirus 2 (SARS-CoV-2) in 2020. This virus exhibits rapid spread among humans, and multiple mutant variants have emerged while spreading. Droplet infection is the major route of SARS-CoV-2 spread; therefore, infection within the respiratory system is most common among patients with COVID-19. The spike protein of SARS-CoV-2 interacts with angiotensin-converting enzyme-2 on host epithelial cells and induces cellular damage. Beyond the direct damage initiated by SARS-CoV-2, a cytokine storm driven by macrophages can be initiated by inflammasomes activated by SARS-CoV-2 entry, the delay in type 1 interferon production [1], macrophage activation induced by proinflammatory cytokines [2,3], Toll-like receptor 4- tumor necrosis factor (TNF) receptor associated factor 6- nuclear factor kappa-light-chain-enhancer of activated B cells (NF-κB) pathway activation induced by oxidized phospholipids [4], and excessive antibody-dependent enhancement mediated by the interaction between SARS-CoV-2 RNA and non-neutralizing IgG3 [5]. Cytokine storm and NOD-, LRR- and pyrin domain-containing protein 3 (NLRP3)-associated inflammasome activation initiated by SARS-CoV-2 induces systemic inflammation and, therefore, dysregulates the cardiovascular and hematologic systems, which creates a state of critical illness during SARS-CoV-2 infection. Since SARS-CoV-2 influences inflammatory dysregulation, the development of adequate adaptive immunity before viral entry, such as that induced by vaccination or an adjunctive strategy, is important for containing SARS-CoV-2 spread and lessening SARS-CoV-2-related critical illness.

The major categories of SARS-CoV-2 vaccines are purified viral components, replication-defective viral vectors carrying pathogen gene(s), and mRNA vaccines. The ChAdOx1 COVID-19 vaccine is categorized as a vaccine derived from a defective adenovirus carrying the full-length codon of the spike protein with a tissue plasminogen activator leader sequence [6]. The protocol for ChAdOx1 injection is administration of 5 × 10^10^ viral particles per dose with two injections administered with an interval of at least 28 days [7]. Multiple variants of SARS-CoV-2 have developed during the COVID-19 pandemic, but ChAdOx1 provides adequate efficacy in preventing the severe issues associated with the different variants, similar to vaccines in other categories [8,9,10]. Vaccine immunogenicity after injection can be assessed by measuring neutralizing antibodies against the receptor for the binding domain (RBD) of the spike protein or by performing an ex vivo interferon-γ enzyme-linked immunospot assay to enumerate antigen-specific T cells. There are two major ways to measure neutralizing antibodies: the absolute amount of a neutralizing binding antibody can be measured by enzyme-linked immunosorbent assay (ELISA) [11,12,13], and the binding capacity of the region containing the binding domain can be measured by the surrogate virus neutralization test [14]. Antibody titers, which are indicative of the humoral response after vaccine injection, have been regarded as the measure for assessing efficacy [15,16], and both the titer of and variation in a neutralizing antibody can predict vulnerability for SARS-CoV-2 infection [17,18]. Beyond the policy of booster administration [19], identifying subjects with a poor response after injection is important for vaccine policy during the COVID-19 pandemic era.

Among people infected with SARS-CoV-2, patients with chronic kidney disease (CKD) are at higher risk for critical illness, along with other comorbidities such as obesity, diabetes mellitus and advanced age [5,6]. The primary etiologies for chronic kidney disease, such as hypertension and diabetes mellitus, are similar to the risk factors for SARS-CoV-2-related critical illness. In addition, the innate or adaptive immune system is usually impaired in CKD patients with varying stages of disease and in patients who underwent kidney transplantation. It has been reported that the efficacy of other vaccines could be impaired in CKD patients because of insufficient production of endogenous erythropoietin, vitamin D deficiency or uremic milieu. Among the different uremic toxins, indoxyl sulfate is a widely discussed protein-bound uremic toxin that suppresses immune cells. Indoxyl sulfate originates from tryptophan, and its interactions with cytotoxic T cells and monocytes have been related to multiple systemic disorders, such as endothelial dysfunction, or shown to induce a pro-inflammatory status in different subjects after infection [20,21,22]. Since vaccine efficacy is highly dependent on the secretion of the cytokine interleukin 17 by helper T cells, the pro-inflammatory status might influence the efficacy of vaccines for SARS-CoV-2 [23].

To date, studies have demonstrated that the efficacy of mRNA vaccines in end stage renal disease (ESRD) patients is hampered [24]. Beyond the well-known factors such as diabetes mellitus and older age, it is unknown whether protein-bound uremic toxins, such as indoxyl sulfate, can influence the efficacy of vaccines based on a replication-defective viral vector carrying a pathogen gene, such as ChAdOx1. The aim of the study was to investigate whether indoxyl sulfate is associated with a poor response to ChAdOx1 in ESRD patients.

## 2. Materials and Methods

### 2.1. Patients and Ethics

This study was conducted at Taipei Tzu Chi Hospital and Cardinal Tien Hospital in New Taipei City, Taiwan, in accordance with the tenets outlined in the Declaration of Helsinki. The study protocol was approved by the Ethics Committees of Human Studies at Taipei Tzu Chi Hospital (09-X-129) and Cardinal Tien Hospital (CTH-110-2-5-045).

### 2.2. Theory and Calculation

#### 2.2.1. Study Design

The study period was from 1 June 2021 to 31 December 2021. The inclusion criteria were as follows: (1) subjects were able to understand written and verbal Chinese or Taiwanese; (2) subjects were a patient or faculty member in the dialysis unit in the network of Taipei Tzu Chi Hospital or Cardinal Tien Hospital; and (3) subjects had received the 1st dose of the ChAdOx1 vaccine. The exclusion criteria were as follows: (1) age < 20 years; (2) under admission status; (3) uncontrolled hypertension, active infection or unstable hemodynamic status; (4) aphasic, illiterate, or unable to write in or understand Chinese or Taiwanese; and (5) previous history of SARS-CoV-2 infection. Since 15 May 2021, the incidence has increased abruptly [25]. Both patients and faculty members in the dialysis unit were eligible to receive the 1st dose of the ChAdOx1 vaccine under the policy of the Central Epidemic Command Center in Taiwan. After obtaining written informed consent from the enrolled participants, the participants were divided into 2 groups based on their clinical characteristics: (1) the control group and (2) the end-stage renal disease group with maintenance hemodialysis (HD group). The control group was derived from the faculty of the dialysis units at Taipei Tzu Chi Hospital or Cardinal Tien Hospital. The members of the faculty were free from end-stage renal disease. The HD group was defined as follows: patients receiving maintenance hemodialysis (three times per week) continuously for >3 months were enrolled. All patients in the HD group received conventional hemodialysis using a high-flux or high-efficiency dialyzer. We obtained written informed consent from the enrolled participants.

Blood and urine samples were obtained. Demographic data such as age, sex and dialysis vintage were obtained from medical records at Taipei Tzu Chi Hospital. Diagnosis of diabetes mellitus was confirmed using medical records. Body weight and height were measured after hemodialysis, and patient body mass index was calculated. Pre-dialytic hematologic and biochemical parameters were obtained at midday (Wednesday or Thursday), within the month after informed consent was obtained; the following parameters were measured: white blood cell count and the percentage of lymphocytes among the total white blood cells. Serum urea levels were recorded pre- and post-dialysis to calculate the single-pool fractional clearance of urea (Kt/V), which serves as the parameter indicating the adequacy of dialysis [26].

#### 2.2.2. Measurement of Indoxyl Sulfate and Antibody Titers after ChAdOx1 Vaccine Injection

The measurement of antibody titers indicative of the humoral response was performed 4 weeks after ChAdOx1 vaccine injection. Fasting blood was drawn from the participants at midday (Wednesday or Thursday) before a dialysis session. Human serum indoxyl sulfate (IS) concentrations were determined using an enzyme-linked immunosorbent assay kit (FineTest, Wuhan, China) according to the manufacturer’s instructions. We used an anti-SARS-CoV-2 antibody immunoglobulin G (IgG) titer serologic assay kit (Spike S1) (Acro Biosystems, Newark, DE, USA) to quantify IgG antibodies in patient plasma. The antibody titer of each sample corresponded to the highest dilution factor that still yielded a positive reading. A value of 10 arbitrary units per milliliter (AU/mL) was considered evidence of a vaccination response. We used a GENLISA™ SARS-CoV-2 (COVID-19) surrogate virus neutralization test (sVNT) ELISA (Krishgen, Biosystems, Cerritos, CA, USA) to detect anti-SARS-CoV-2 antibodies suppressing the interaction between the RBD of the viral spike (S) glycoprotein and the angiotensin-converting enzyme 2 (ACE2) protein on the surface of cells. Data are presented as the inhibition rate (%).

#### 2.2.3. Statistics

Continuous variables are presented as the means and standard deviations. Categorical values are expressed as the frequency of the count and the percentage. Kruskal–Wallis one-way analysis of variance was used for comparisons of continuous variables among the four groups, including biochemical and laboratory data. The chi-square test was used to analyze the associations between categorical variables. To evaluate the dose-dependent effect of IS, we divided the ESRD patients into quartiles based on the IS concentration. The *p* for trend was used to compare the anti-S1 IgG titers and sVNT neutralization titers from the 1st quartile to the 4th quartile. To evaluate the variables associated with indoxyl sulfate and demographic factors, we used univariate and multivariate linear regression models to determine the regression coefficients with 95% confidence intervals (CIs) for the clinical variables. All potential variables (*p* < 0.05) identified in the univariate linear regression analysis were entered into multivariate linear regression models with forward methods. All statistical analyses were performed using STATA 17.0 (STATA Corp, College Station, TX, USA) and SAS 9.4 (SAS Institute Inc., Cary, NC, USA). A two-sided *p* value of <0.05 was considered statistically significant.

## 3. Results

### 3.1. The Concentration of Anti-S1 IgG Was Lower in Hemodialysis Patients Than in the Control Group

Figure 1 illustrates the serum IgG antibody titer (Panel A) and sVNT value (Panel B) against the S1 subunit of the spike protein of SARS-CoV-2 in the hemodialysis patients (*n* = 358) and control group (*n* = 59). The serum concentrations of anti-S1 IgG were 272 ± 1726 AU/mL and 2111 ± 4424 AU/mL in the hemodialysis and control groups, respectively (*p* < 0.05). The sVNT values were 26.8 ± 21.1% and 54.0 ± 20.2% in the hemodialysis and control groups, respectively (*p* < 0.05).

### 3.2. The Humoral Response against SARS-CoV-2 Was Lower in Hemodialysis Patients

Figure 2 illustrated the Spearman’s rank correlation coefficient between the sVNT and the anti-S1 antibody titer of the hemodialysis participants and the control group. The correlation between these two indicators was 0.31 in hemodialysis participants (*p* < 0.05) and 0.42 in control group (*p* < 0.05)

Figure 3 illustrates the kernel density estimations for the anti-S1 antibody titer (Panel A) and sVNT value (Panel B) between hemodialysis patients and the control group. Both the kernel density estimations showed that the anti-S1 IgG titer and sVNT value were significantly lower in the hemodialysis patients than in the control group.

### 3.3. The Higher Concentration of IS Was Associated with the Lower Humoral Response after Vaccination with ChAdOx1

To compare the effect of the IS concentration on the humoral response after vaccination, we firstly performed the Spearman correlation between serum IS concentration and the IgG titer (Spearman’s rho = −0.11, *p* = 0.049 for anti-S IgG; Spearman’s rho = −0.02, *p* = 0.65 for percentage of neutralization). Then we divided the hemodialysis patients into four groups based on the circulating IS concentration quartile: the first quartile (Q1), 95.03 μg/dL or less; the second quartile (Q2), 95.03 to 147.25 μg/dL; the third quartile (Q3), 147.255 to 211.59 μg/dL; and the 4th quartile (Q4), 211.59 μg/dL or greater. Table 1 demonstrates the baseline demographics and relevant clinical characteristics of the four groups. There were no significant differences in the parameters except the serum IS concentration and percentage of peripheral blood lymphocytes < 20%.

The anti-S1 IgG titers were 146.7 ± 254.7 AU/mL, 134.9 ± 308.8 AU/mL, 101.1 ± 195.1 AU/mL, and 176.6 ± 382.4 AU/mL from the first to fourth quartiles (*p* = 0.54). The sVNT values were 28.22 ± 22.22%, 27.06 ± 20.05%, 27.5 ± 21%, and 27.4 ± 20% from the first to fourth quartiles (*p* = 0.79). When comparing the IgG titer and the sVNT between groups, there was no difference when comparing the concentration by one way analysis of variance. When performing the *p* for trend, the decrease in IgG (by median value) and the sVNT (by mean value) were noted between groups. The distribution is illustrated in Figure 4. The anti-S1 IgG titer (by median value) showed a decreasing trend from the first to fourth quartiles (Panel A, *p* < 0.05). The sVNT results (by mean value) also showed a decreasing trend from the first to fourth quartiles (Panel B, *p* < 0.05).

Table 2 demonstrates the factors linked to the titer of anti-S1 IgG. In the crude model, higher IS concentrations (Q2 to Q4) were associated with lower circulating anti-S1 antibody levels compared with the lower IS concentration in the Q1 group (reference). Elderly age (155.9, 95% CI for regression coefficients: 107.4 to 204.4, *p* < 0.01), male sex (105.3, 95% CI: 63.4 to 147.3, *p* < 0.01), diabetes mellitus (108.2, 95% CI: 66.2 to 150.2, *p* < 0.01), dialysis vintage (102.6, 95% CI: 46.2 to 159.0, *p* < 0.01), and the percentage of lymphocytes (92.9, 95% CI: 51.4 to 134.4, *p* < 0.01) were also associated with the anti-S1 IgG titer. In addition, the serum concentration of IS was associated with the titer of anti-S1 IgG in a dose-dependent manner (*p* for trend < 0.001). After adjusting for age, sex, diabetes mellitus status, and lymphocyte percentage, the concentration of IS (Q2: −119.1, 95% CI: −207.8 to −30.4, *p* < 0.01; Q3: −225.1, 95% CI: −340.2 to −110.1, *p* < 0.01; Q4: −448.2 95% CI: −622.0 to −274.5, *p* < 0.01) still negatively influenced the anti-S1 IgG titer in a dose-dependent manner (*p* for trend < 0.01).

Table 3 demonstrates the factors affecting the sVNT values of hemodialysis patients. The serum IS concentration (Q2: 1.7, 95% CI: −6.8 to 10.3, *p* = 0.69; Q3: −8.8, 95% CI: −19.4 to 1.9, *p* = 0.11; Q4: −27.2, 95% CI: −42.4 to −12.0, *p* < 0.01), older age (30.1, 95% CI: 24.6 to 35.6, *p* < 0.01), male sex (25.4, 95% CI: 20.8 to 30.1, *p* < 0.01), diabetes mellitus (27.7, 95% CI: 23.3 to 32.1, *p* < 0.01), dialysis vintage (27.3, 95% CI: 20.8 to 33.9, *p* < 0.01), and lymphocyte percentage (24.0, 95% CI: 19.4 to 18.5, *p* < 0.01) were significantly associated with the percentage calculated with the sVNT. Additionally, the serum concentration of IS inversely influenced sVNT values in a dose-dependent manner (*p* for trend < 0.001). After adjusting for age, sex, diabetes mellitus, and the percentage of lymphocytes, the higher concentration of IS (Q2: 2.2, 95% CI: −6.2 to 10.7, *p* = 0.61; Q3: −5.8, 95% CI: −16.6 to 5.1, *p* = 0.29; Q4: −21.2 95% CI: −37.8 to −4.5, *p* = 0.013) remained associated with lower sVNT values in a dose-dependent manner (*p* for trend < 0.001).

## 4. Discussion

Our study showed that ESRD patients receiving maintenance hemodialysis had a lower humoral response after the first dose of ChAdOx1 than the control group. In the ESRD patients, the higher serum concentration of indoxyl sulfate was associated with a lower humoral response.

To our knowledge, our report is the first to report the interaction between indoxyl sulfate and humoral response after ChAdOx1 injection in patients with ESRD. During the COVID-19 pandemic, the efficacy of SARS-CoV-2 vaccines has been evaluated by measuring both the cellular and humoral responses [15,27]. From a study in the United Kingdom, neutralizing antibodies and helper T cells developed mostly 3–4 weeks after SARS-CoV-2 infection [28]. This timeline for the development of immunity made it rational for our study to measure titers at 4 weeks after the first injection of the ChAdOx1 vaccine. In ESRD patients, the humoral response after COVID-19 infection was similar to that in healthy subjects. Therefore, the measurement of anti-S1 immunoglobulin in ESRD patients was a valid way to assess the immunogenicity of SARS-CoV-2 vaccine [29]. Ebinger et al. demonstrated that an anti-S1 IgG titer higher than 4160 AU/mL might be a cut-off value for the efficacy of mRNA vaccines by comparison with post-COVID infection subjects [30]. mRNA vaccines, in comparison with ChAdOx1, induce a stronger humoral response after the first injection in general subjects. The control group in our observational study also developed a relatively higher titer of anti-S1 antibodies. In previous studies on the efficacy of SARS-CoV-2 vaccines in ESRD patients, mRNA vaccines were mostly discussed [31,32,33]. The anti-S1 IgG titer was lower in ESRD patients. From the meta-analysis by Chang et al., the immunogenicity rate was 41% in ESRD patients after the first injection, and diabetes mellitus was an important factor associated with a poor immune response after vaccine injection [34]. In our results, diabetes mellitus was not found to be a contributing factor after adjustment for other confounding factors. Perhaps an observational study with a larger sample size might further validate the effect of diabetes mellitus after ChAdOx1 infection in ESRD patients.

Our results demonstrated that the concentration of indoxyl sulfate, along with older age and a lower lymphocyte percentage, was associated with a weaker humoral response after vaccine injection. Indoxyl sulfate has been reported to influence hematopoietic progenitor cells from the bone marrow in CKD subjects with endothelial injury [35]. Among subjects receiving SARS-CoV2 vaccine injection, fewer CD8+ T cells producing interferon alpha and TNF-αwere found in the subjects without previous infection, and CD4+ cells with interleukin 2 expression were enriched as the major immune cells after vaccine injection [28]. In the study by Kim et al., indoxyl sulfate was shown to induce chronic inflammation in ESRD patients. Endothelial cells injured by indoxyl sulfate increased the expression of CX3C chemokine receptor 1 (CX3CL1), which interacted with CX3CR1 on CD4^+^CV28^−^ T cells capable of cytotoxic function by releasing tumor necrosis factor alpha [20]. Xiang et al. also demonstrated that the RNA transcripts of TNF α and TNF-alpha and INF-γ were increased in CKD patients and that the aryl hydrocarbon receptor gene was upregulated along with the increases in pro-inflammatory cytokines [36]. Macrophages, as antigen-presenting cells, are also targets of indoxyl sulfate [37]. Nakano et al. reported that indoxyl sulfate activated the pro-inflammatory status of macrophages by entry via organic anion transporter (OATP2B1) and that notch signaling was activated within macrophages after indoxyl sulfate exposure [37]. The differentiation of osteoclasts, which originate from macrophages, was also abated by a high dose of indoxyl sulfate via activation of the aryl hydrocarbon receptor and its regulation of the expression of the osteoclast precursor nuclear factor of activated T cells, cytoplasmic 1 [38]. Indoxyl sulfate influences cellular function by increasing intracellular oxidative stress or by modulating gene expression associated with the aryl hydrocarbon receptor [39]. Excessive oxidative stress induces apoptosis and hypo-responsiveness in T cells [40]. Indoxyl sulfate might contribute to chronic T cell exhaustion in CKD/ESRD patients by causing chronic exposure to proinflammatory factors [41]. In our study, a higher concentration of indoxyl sulfate, along with a lower lymphocyte count, was associated with a lower anti-S1 IgG titer. The studies focusing on the protein bound uremic toxin and hypo-responsiveness of vaccine are few. Further studies are needed focusing on indoxyl sulfate mediated disturbance on the maturation, differentiation and viability of the T cells. A strategy to lower the systemic level of indoxyl sulfate, such as administration of AST-120 [39,42] or other conjunctive agents, such as resveratrol [43], might potentiate the efficacy of vaccines against SARS-CoV-2.

Recently, the study by Rincon-Arevalo et al. demonstrated that B cell maturation was impaired in ESRD patients and that this impairment was associated with a poor humoral response in ESRD patients after mRNA vaccination [32]. In the control group, antigen-specific B cells were identified in the plasmablast or post-switch memory B cell stage. However, antigen-specific B cells were identified to be in the pre-switch or naïve B cell stage in ESRD patients. In the study, the age of the control group was younger than that of the ESRD group. It has been reported that the B cell maturation profile can be impacted by age. Ciocca et al. illustrated that CD27^dull^ memory B cells, which are cells bridging innate and adaptive immunity after exposure to new pathogens, were reduced in elderly individuals [44]. Ju et al. also showed that antibody diversity was decreased in elderly individuals receiving seasonal influenza vaccine injection. Flow cytometry demonstrated that clonality after vaccine injection was increased in elderly individuals compared with younger responders, but diversity among the clones was impaired in the elderly individuals [45]. Class switching might be impaired in the elderly and, therefore, might influence the efficacy of vaccines against new pathogens [46,47]. It is known that the prevalence and incidence of CKD/ESRD are increased in the elderly [48]. In our results, older age contributed to a weaker immune response in ESRD patients, as measured by both the anti-S1 IgG titer and sVNT values.

Our study had several limitations. First, all the subjects included in the study received only one dose of the ChAdOx1 vaccine. Although anti-S1 antibodies developed in the control group, the ESRD patients in our study might not be regarded as non-responders after vaccine injection. From studies related to mRNA vaccines, a higher vaccine dose or an extra booster might elicit immunogenicity in ESRD patients receiving hemodialysis [49]. The titer after booster administration should be evaluated. Second, our study measured only anti-S1 Ab titers and sVNT values. As discussed above, the profiles of T cells and B cells might influence the development of antibodies [50]. Additionally, we performed the *p* for trend based on the mean value of sVNT and median value of IgG. In Figure 1, we illustrated that there was correlation between the sVNT and IgG titer. Discrepancy between the titers and percentage of neutralization were noted, and a larger sample size or further studies focusing on the immunogenicity might be needed. Third, the influence of dialysis was not measured. In the study by Speer et al., dialysis adequacy was shown to influence antibody development [51]. Dialysis also influences the profile of lymphocytes [52]. However, more than 80% of all the subjects received adequate dialysis according to the definition of the KDOQI clinical guidelines [53]. A larger sample size might be needed to evaluate the effect of dialysis adequacy. Fourth, our result demonstrated the association between the concentration of IS and the markers indicating the humoral response after SARS-CoV2 vaccine injection. However, the causality between the IS and the development of IS might need further trials or even in vivo studies to clarify. The IS concentration in dialysis patients was much higher than other CKD patients [54]. Therefore, to enroll patients with different CKD stage and correlating the serum IS with antibody titer might provide the causality. Finally, vaccine efficacy related to the clinical outcome of SARS-CoV-2 infection, such as disease-related mortality or comorbidities, could not be measured due to the relatively low incidence of COVID-19 in Taiwan [55,56].

## 5. Conclusions

In comparison with control group, patients with ESRD had weaker humoral immunity after the first ChAdOx1 vaccine injection. The higher concentration of indoxyl sulfate altered influenced the development of anti-S1 antibodies and sVNT values in ESRD patients.

## Figures and Tables

**Figure 1 vaccines-10-01378-f001:**
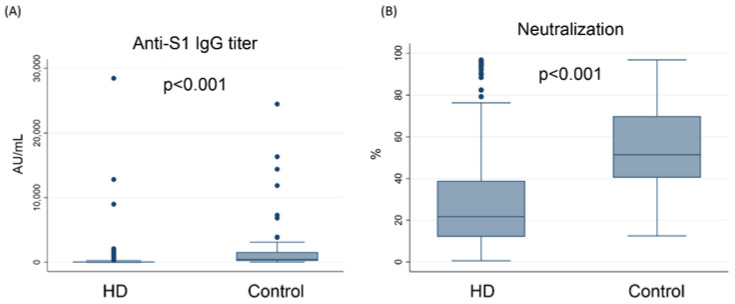
Anti-S1 IgG concentrations (**A**), and sVNT-measured neutralization (**B**), between control group and hemodialysis patients. Abbreviations: AU, arbitrary unit; HD, hemodialysis; S1, the S1 sub-unit of the spike protein; sVNT, surrogate virus neutralization test.

**Figure 2 vaccines-10-01378-f002:**
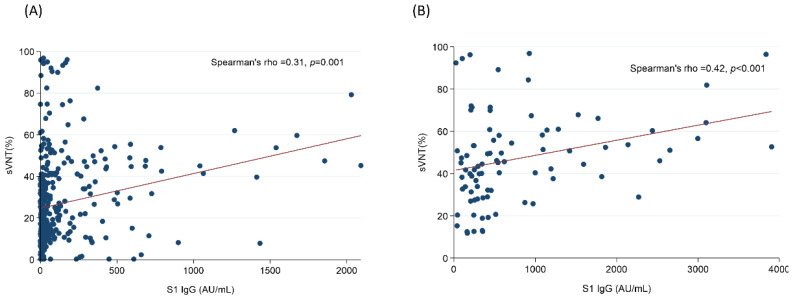
The correlation coefficient between the anti-S1 IgG and sVNT in the hemodialysis (**A**) group, and the control (**B**) group.

**Figure 3 vaccines-10-01378-f003:**
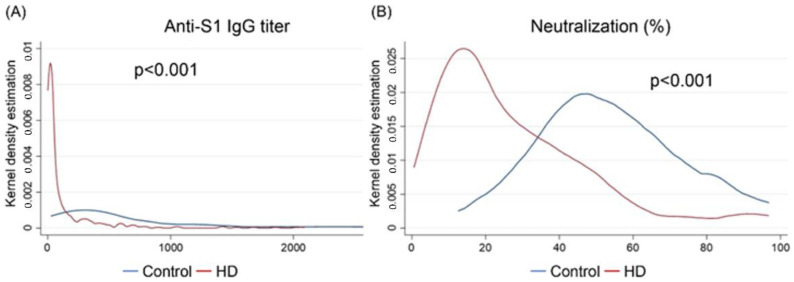
Kernel density estimations for anti-S1 IgG titers (**A**), and sVNT-measured neutralization (**B**), between the control group and hemodialysis patients. Abbreviations: AU, arbitrary unit; HD, hemodialysis; S1, the S1 sub-unit of the spike protein; sVNT, surrogate virus neutralization test.

**Figure 4 vaccines-10-01378-f004:**
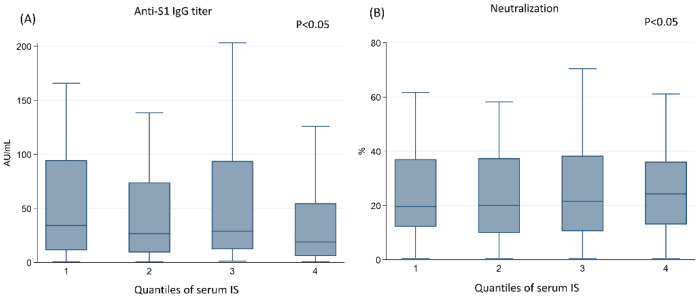
Comparisons of the (**A**) anti-S1 IgG titer and (**B**) sVNT-measured neutralization based on the quartiles of the serum indoxyl sulfate concentration by *p* for trend. Abbreviations: AU, arbitrary unit; IS, indoxyl sulfate; S1, the S1 sub-unit of the spike protein; sVNT, surrogate virus neutralization test.

**Table 1 vaccines-10-01378-t001:** Demographics of the hemodialysis patients grouped based on the concentration of indoxyl sulfate.

Characteristics	Indoxyl Sulfate Concentrations				*p* Value
	Quartile 1 (<95.03)	Quartile 2 (95.03–147.255)	Quartile 3 (147.255–211.59)	Quartile 4 (≥211.59)	
Patient number	91	88	90	89	
Age group, years					0.15
<65	45 (49.5)	56 (63.6)	58 (64.4)	52 (58.4)	
≥65	46 (50.5)	32 (36.4)	32 (35.6)	37 (41.6)	
Mean (sd)	67 (11.1)	65.2 (12)	67.4 (15)	67.4 (11.7)	0.59
Sex, n (%)					0.20
Male	57 (62.6)	42 (47.7)	47 (52.2)	52 (58.4)	
Female	34 (37.4)	46 (52.3)	43 (47.8)	37 (41.6)	
Diabetes mellitus, n (%)	43 (47.3)	48 (54.5)	51 (56.7)	42 (47.2)	0.46
WBC count, per 1000/μL					
Mean (sd)	6 (1.6)	6.2 (1.8)	6.6 (2.2)	6.3 (2)	0.24
Median (IQR)	6 (5–7)	5.9 (4.9–7.3)	6.5 (5.1–7.8)	5.8 (5–7.6)	0.39
≥8.2	11 (12.1)	10 (11.4)	18 (20)	15 (16.9)	0.32
Lymphocyte, %					
Mean (sd)	22.6 (8.1)	22.6 (6.6)	21.9 (8.2)	22.1 (8)	0.64
Median (IQR)	21.7 (17.6–26.2)	22.4 (18–27.2)	21.8 (16.7–25.8)	21.8 (17.3–26.4)	0.77
<20	44 (48.4)	43 (48.9)	58 (64.4)	56 (62.9)	0.04
Kt/V					
Mean (sd)	1.6 (0.3)	1.6 (0.3)	1.5 (0.3)	1.6 (0.3)	0.91
Median (IQR)	1.5 (1.4–1.8)	1.6 (1.4–1.8)	1.5 (1.3–1.7)	1.6 (1.4–1.8)	0.21
≥1.2	84 (92.3)	82 (93.2)	79 (87.8)	84 (94.4)	0.39
Dialysis vintage, years					
Mean (sd)	4.9 (5.6)	5.4 (5.8)	3.7 (3.8)	5.4 (6.9)	0.92
Median (IQR)	2.9 (1.4–6.4)	3.5 (2.2–6.7)	2.8 (1.5–4.4)	3.3 (1.4–5.9)	0.11
≥5 years	44 (48.4)	53 (60.2)	36 (40)	46 (51.7)	0.06
Indoxyl sulfate					
Mean (sd)	74 (18.6)	119.3 (14.6)	176 (17.2)	381.2 (443.3)	<0.001
Median (IQR)	78.8 (65.1–87.7)	119.9 (106.6–131.2)	172.8 (162.5–190.6)	273.6 (249.1–357.4)	<0.01
PCS					
Mean (sd)	20.7 (20.1)	24.1 (25.6)	32.3 (72.3)	81.4 (454.1)	0.08
Median (IQR)	12.4 (4.6–31.7)	9.6 (4.7–34.1)	10.5 (4.9–31.5)	9.5 (4.4–29.9)	0.96
Anti-S1 IgG, AU/mL					
Mean (sd)	146.7 (254.7)	134.9 (308.8)	101 (195.1)	176.6 (382.4)	0.54
Median (IQR)	37.8 (14–126.4)	29.8 (12.2–115.2)	29.2 (12.3–102.7)	27.6 (12–132.1)	0.74
sVNT (% neutralization)					
Mean (sd)	28.2 (22.2)	27.6 (20.5)	27.5 (21)	27.4 (20)	0.79
Median (IQR)	20.5 (12.5–38.4)	23.8 (11.3–40.7)	21.9 (10.7–39.3)	21.8 (12.7–35.8)	0.98

**Table 2 vaccines-10-01378-t002:** Associations between anti-S1 IgG and indoxyl sulfate levels at 4 weeks after the 1st dose of the ChAdOx1 vaccine in patients undergoing hemodialysis.

	Crude Model		Adjusted Model ^a^	
	Regression Coefficients (95% Confidence Intervals)	*p* Value	Regression Coefficients (95% Confidence Intervals)	*p* Value
Indoxyl sulfate				
Quartile 1	reference		reference	
Quartile 2	−110.5 (−200.1 to −20.8)	0.016	−119.1 (−207.8 to −30.4)	0.009
Quartile 3	−189.4 (−301.9 to −77.0)	0.001	−225.1 (−340.2 to −110.1)	<0.001
Quartile 4	−352.4 (−511.9 to −192.9)	<0.001	−448.2 (−622.0 to −274.5)	<0.001
*p* for trend		<0.001		<0.001
Age ≥ 65 vs. age < 65	155.9 (107.4 to 204.4)	<0.001	61.5 (5.3 to 117.7)	0.032
Male vs. female sex	105.3 (63.4 to 147.3)	<0.001	−34.8 (−90.9 to 21.3)	0.22
Diabetes mellitus	108.2 (66.2 to 150.2)	<0.001	−45.1 (−101.4 to 11.2)	0.12
Dialysis vintage ≥ 5 vs. <5 years	102.6 (46.2 to 159.0)	<0.001	−37.1 (−97.7 to 23.5)	0.23
% Lymphocyte < 20% vs. ≥20%	92.9 (51.4 to 134.4)	<0.001	−66.0 (−124.1 to −7.8)	0.026

^a^ Adjusted for age, sex, diabetes, Kt/v, dialysis vintage, and percentage of lymphocytes.

**Table 3 vaccines-10-01378-t003:** Contributing factors in sVNT-measured neutralization at 4 weeks after administration of the 1st dose of the ChAdOx1 vaccine in patients undergoing hemodialysis.

	Crude Model		Adjusted Model ^a^	
	Regression Coefficients (95% Confidence Intervals)	*p* Value	Regression Coefficients (95% Confidence Intervals)	*p* Value
Indoxyl sulfate				
Quartile 1	reference		reference	
Quartile 2	1.7 (−6.8 to 10.3)	0.69	2.2 (−6.2 to 10.7)	0.61
Quartile 3	−8.8 (−19.4 to 1.9)	0.11	−5.8 (−16.6 to 5.1)	0.29
Quartile 4	−27.2 (−42.4 to −12.0)	<0.001	−21.2 (−37.8 to −4.5)	0.013
*p* for trend		<0.001		<0.001
Age ≥ 65 vs. age < 65	30.1 (24.6 to 35.6)	<0.001	8.8 (3.5 to 14.1)	0.001
Male vs. female	25.4 (20.8 to 30.1)	<0.001	0.9 (−4.4 to 6.2)	0.74
Diabetic	27.7 (23.2 to 32.1)	<0.001	3.4 (−1.9 to 8.7)	0.21
Dialysis vintage ≥ 5 vs. <5 years	27.3 (20.8 to 33.9)	<0.001	3.2 (−2.5 to 8.9)	0.27
% Lymphocyte < 20% vs. ≥20%	24.0 (19.4 to 28.7)	<0.001	−3.2 (−8.7 to 2.3)	0.25

^a^ Adjusted for age, sex, diabetes, Kt/v, dialysis vintage, and percentage of lymphocytes.

## Data Availability

Not applicable.

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
