# Peer review of "Indoxyl Sulfate Alters the Humoral Response of the ChAdOx1 COVID-19 Vaccine in Hemodialysis Patients"

_vaccines, 2022, doi:10.3390/vaccines10091378_

Round 1

Reviewer 1 Report

This paper demonstrated immunity response of patients with ESRD were affected by the concentration of indoxyl sulfate. It reads clear and written well. The topic is very interesting. More detailed and comprehensive studies regarding the molecular mechanism are necessary to further strengthen the claims that were made and demonstrate the significance of this work. A perspective should also be included in the conclusion. The paper can be accepted with minor revision. 

Author Response

Thank you for your valuable comment. As the reviewer’s suggestion, future work are needed to demonstrate the significance of the study. After reviewing the manuscript, we added the statement as follows:
1. our report is the first to report the interaction between IS and humoral response after ChAdOx1 injection in patients with ESRD (line 263-264)

  1. The studies focusing on the protein bound uremic toxin and hypo-responsiveness of vaccine are few. Further studies are needed focusing on indoxyl sulfate mediated disturbance on the maturation, differentiation and viability of the T cells. A strategy to lower the systemic level of indoxyl sulfate, such as administration of AST-120 [39,42] or other conjunctive agents, such as resveratrol [43], might potentiate the efficacy of vaccines against SARS-CoV-2.(line 312-317)

Reviewer 2 Report

Indoxyl sulfate impairs the humoral response of the ChAdOx1 COVID-19 vaccine in hemodialysis patients

This is an interesting topic and the authors have taken the necessary steps to make sure the objectives of the study achieved.

The paper is very well written. I have some minor comments:

Abstract

Abstract: Background and aims: Vaccination for severe acute respiratory syndrome coronavirus 2(SARS-CoV-2) is strongly recommended. The efficacy of SARS-CoV-2 vaccine for patients with end-stage renal disease is low. It is unknown whether indoxyl sulfate (IS) impairs the efficacy of vaccines for SARS-CoV-2. Materials and Methods: During June 1st, 2021, to December 31st, 2021, hemodialysis patients (n=358) and control group (n=59) were eligible to receive the 1st dose of the ChAdOx1 COVID-19 vaccine. Titer measurements indicative of the humoral response (anti-S1 IgG and surrogate virus neutralization test [sVNT] results) and indoxyl sulfate concentration measurement were per-formed 4 weeks after ChAdOx1 vaccine injection. Results: The serum concentrations of anti-S1 IgG were 272 ± 1726 AU/mL and 2111 ± 4424 AU/mL in hemodialysis patients and control group (p<0.05). The sVNT values were 26.8±21.1% and 54.0±20.2% in the hemodialysis and control groups (p<0.05). There was a decreasing trend for the anti-S1 IgG titer from the lowest to highest quartile of IS (p<0.001). The concentration of IS was negatively associated with sVNT results (p for trend<0.001). Conclusion: Hemodialysis patients had weaker humoral immunity after the first dose of the ChAdOx1 vaccine. The concentration of IS negatively influenced the development of anti-S1 an-tibodies and sVNT-measured neutralization..

Comment: Background info about IS should be given to give context of the description

METHOD

Comment

The randomisation should be further elaborated

Please indicate which part of the  CONSORT check list was not adhered to?

Figure 2. The correlation between the anti-S1 IgG and sVNT in the hemodialysis (A) group and the 190 control (B) group.

Comment: This correlation does not appear to be appropriate

 the resolution of all figures should be improved.

Reviewer 3 Report

Review of Manuscript “Indoxyl sulfate impairs the humoral response of the ChAdOx1 COVID-19 vaccine in hemodialysis patients“ by Yi-Chou Hou et al..

It is known that the efficacy of vaccination against SARS-CoV-2 in patients with severe renal disease is low. In the present manuscript the authors examine the formation of anti SARS-CoV-2 Spike protein antibodies (ELISA and surrogate virus neutralization test) after a single application of a vaccine based on recombinant chimpanzee adenovirus (ChAdOx1) expressing the Spike protein. As expected, they find lower titers of total anti-Spike IgG and neutralizing antibodies in end-stage renal disease patients as compared to a control group of healthy persons.

By division of hemodialysis patients into four groups based on on the concentration of circulating indoxyl sulfate (IS), an uremic toxin often elevated in chronic kidney disease patients, the authors then postulate the main claim of the paper, which is the negative association of the vaccination response with IS concentration. However, the primary data shown regarding this claim is seriously flawed, so that it is not possible to evaluate its correctness. In the text (line 212) a mean value 728.5 ± 3504.8 AU/ml is stated for the anti-S1 titers of the second quartile (Q2) of patients. However, in the correponding diagram (fig. 4A), no single value above 700 AU/ml can be recognized with most values well below 100 AU/ml! So it is quite hard to comprehend, how the mean value of 728.5 AU/ml, which is also given in table 1) was calculated! There also seem to be major discrepancies between the mean values (146.7, 728.5, 101 and 176.6 AU/ml for the first to fourth quartile, respectively) and the correponding medians (37.8, 29.8, 29.2 and 27.6 AU/ml, respectively) for the anti-S1 IgG titers in table 1, even if one considers that the majority of titers are rather low. Furthermore, as the authors also confess, there was no significant negative correlation between the titers of neutralizing antibodies and IS concentration.          

Regarding the extensive discussion in the corresponding section of the possible molecular mechanisms regarding a link between increased IS concentrations and a reduced ChAdOx1 COVIS-19 vaccine response, it must be stated that even an association between these parameters (which is also highly questionable, see section above) does not implicate any causality.

Round 2

Reviewer 3 Report

In my view, the authors of the study have addressed my concerns of the original manuscript in a very superficial manner. First of all, the attached letter of reply (word document) seems to have gaps. The explanation of the apparent discrepancy between the data in the text (both in the text and table 1!) and in the corresponding graph seems very questionable to me. These discrepancies should have been quite obvious when checking the final manuscript and raise at least some concerns about the data analysis. Furthermore, the authors do not even address my argument that there is no significant negative correlation between the titers of neutralizing antibodies, the second important parameter in addition to the total IgG concentration, and the IS concentration.

Author Response

Reviewer comment:
In my view, the authors of the study have addressed my concerns of the original manuscript in a very superficial manner. First of all, the attached letter of reply (word document) seems to have gaps. The explanation of the apparent discrepancy between the data in the text (both in the text and table 1!) and in the corresponding graph seems very questionable to me. These discrepancies should have been quite obvious when checking the final manuscript and raise at least some concerns about the data analysis. Furthermore, the authors do not even address my argument that there is no significant negative correlation between the titers of neutralizing antibodies, the second important parameter in addition to the total IgG concentration, and the IS concentration.

ANS:

Thank you for your critical comment on the manuscript. We appreciate the reviewer’s comment, which could improve the quality of the manuscript and the study. The following statement is the response from our study group:
1. As the reviewer’s comment, there is discrepancy between the mean and the median of IgG and sVNT between groups according to the IS concentration. There was no difference when comparing the concentration directly between groups by one way analysis of variance. When performing the p for trend, the decrease in IgG (by median value) and the sVNT( by mean value) was noted between group. Based on the finding, we performed further analysis. We added the following statement in the associated section:
a. Line 221-226:
When comparing the IgG titer and the sVNT between group, there was no difference when comparing the concentration by one way analysis of variance. When performing the p for trend, the decrease in IgG (by median value) and the sVNT( by mean value) was noted between group. The distribution is illustrated in Figure 4. The anti-S1 IgG titer(by median value) showed a decreasing trend from the 1st to 4th quartiles (Panel A, p<0.05). The sVNT results (by mean value) also showed a decreasing trend from the 1st to 4th quartiles (Panel B, p<0.05).
b. line 349-353:
Besides, we performed the p for trend based on the mean value of sVNT and median value of IgG. In figure 1, we illustrated that there was correlation between the sVNT and IgG titer. Discrepancy between the titers and percentage of neutralization were noted, and larger sample size or further studies focusing on the immunogenicity might be needed.

  1. The reviewer suggested to correlate the IS concentration with the IgG titer or sVNT. As the reviewer’s suggestion, we performed the correlation. We added the statement in the line 204-208. We removed the word negatively and used lower instead to match our result. Besides, we add our comment in the limitation as follows:
    a. line 33:
    The patients with higher concentration of IS had lower sVNT (p for trend<0.001).
  2. line 36:
    Higher concentration of IS altered the development of anti-S1 antibodies and sVNT-measured neutralization.
  3. line 204:
    3.3. The higher concentration of IS was associated with the lower humoral response after vaccination with ChAdOx1
  4. line 206-209:
    To compare the effect of the IS concentration on the humoral response after vaccination, we firstly performed the Spearman correlation between serum IS concentration and the IgG titer (Spearman’s rho=-0.11, p=0.049 for anti-S IgG; Spearman’s rho=-0.02, p=0.65 for percentage of neutralization)
  5. line 234:
    higher IS concentrations (Q2 to Q4) were associated with lower circulating anti-S1 antibody levels compared with the lower IS concentration in the Q1 group (reference)

f.:line 257:
the higher concentration of IS (Q2: 2.2, 95% CI: -6.2 to 10.7, p=0.61; Q3: -5.8, 95% CI: -16.6 to 5.1, p=0.29; Q4: -21.2 95% CI: -37.8 to -4.5, p=0.013) remained associated with lower

g.:line 369:
The higher concentration of indoxyl sulfate altered influenced the development of an-ti-S1 antibodies and sVNT values in ESRD patients.

h.line 361-364:
The IS concentration in dialysis patients was much higher than other CKD patients [56]. Therefore, to enroll patients with different CKD stage and correlating the serum IS with antibody titer might provide the causality

Round 3

Reviewer 3 Report

Review of second revised version of Manuscript “Indoxyl sulfate alters the humoral response of the ChAdOx1 COVID-19 vaccine in hemodialysis patients“ by Yi-Chou Hou et al..

The authors of the study have now addressed my concerns of the original and first revised versions of the manuscript in a in a reasonably satisfactory manner, which has led to a significant improvement of the manuscript.

Author Response

Thank you for your comment. We've made the adjustment accordingly.